# SUPERVISED REINFORCEMENT LEARNING: FROM EXPERT TRAJECTORIES TO STEP-WISE REASONING

**Yihe Deng**[2][*][†]  **I-Hung Hsu**[1][*]  **Jun Yan**[1]  **Zifeng Wang**[1]  **Rujun Han**[1]  **Gufeng Zhang**[3]
**Yanfei Chen**[1]  **Wei Wang**[2]  **Tomas Pfister**[1]  **Chen-Yu Lee**[1]

[1]Google Cloud AI Research  [2]UCLA  [3]Google Cloud

## ABSTRACT

Large Language Models (LLMs) often struggle with problems that require multi-step reasoning. For small-scale open-source models, Reinforcement Learning with Verifiable Rewards (RLVR) fails when correct solutions are rarely sampled even after many attempts, while Supervised Fine-Tuning (SFT) tends to overfit long demonstrations through rigid token-by-token imitation. To address this gap, we propose Supervised Reinforcement Learning (SRL), a framework that reformulates problem solving as generating a sequence of logical "actions". SRL trains the model to generate an internal reasoning monologue before committing to each action. It provides smoother rewards based on the similarity between the model's actions and expert actions extracted from the SFT dataset in a step-wise manner. This supervision offers richer learning signals even when all rollouts are incorrect, while encouraging flexible reasoning guided by expert demonstrations. As a result, SRL enables small models to learn challenging problems previously unlearnable by SFT or RLVR. Moreover, initializing training with SRL before refining with RLVR yields the strongest overall performance. Beyond reasoning benchmarks, SRL generalizes effectively to agentic software engineering tasks, establishing it as a robust and versatile training framework for reasoning-oriented LLMs.

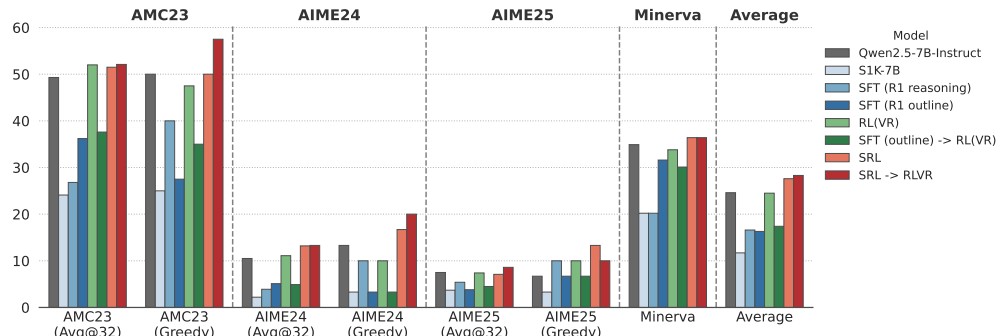

Figure 1: Performance of our method (SRL) against baselines on math reasoning benchmarks, with all models trained on the challenging s1k dataset (Muennighoff et al., 2025). Our key observations are: (1) Directly applying SFT on this dataset leads to performance degradation compared to the base model. (2) While RLVR can improve generalization over SFT, the gains are marginal. (3) Our proposed SRL method substantially outperforms these baselines, and the SRL → RLVR pipeline achieves the highest performance, overcoming the challenges of training on difficult data.

## 1 INTRODUCTION

Large Language Models (LLMs) have demonstrated remarkable versatility across diverse domains, ranging from mathematical problem-solving (Wang et al., 2025) and search-result integration (Jin

---

\* Equal contribution.   † This work was done while the author interned at Google Cloud AI Research.
Correspondence: yihedeng@g.ucla.edu, ihung.hsu@gmail.com, chenyulee@google.com

et al., 2025; Hsu et al., 2024) to tool-use (Yin et al., 2025; Qian et al., 2025) and long-horizon agentic interaction (Xie et al., 2024; Li et al., 2025c; Tan et al., 2025). A pivotal recent advancement involves the use of reinforcement learning (RL) to bolster complex reasoning capabilities (Shao et al., 2024; Ahmadian et al., 2024; Lambert et al., 2024). By optimizing models through reward signals tied to verifiable outcomes—such as final answer accuracy—RL provides a scalable framework for eliciting sophisticated problem-solving behaviors, including self-reflection (Guo et al., 2025; Xie et al., 2025).

The effectiveness of these outcome-based RL methods fundamentally depends on the policy model's ability to discover correct solutions within a limited rollout budget (Brown et al., 2024). However, given practical computational constraints, this learning paradigm struggles on challenging problems from the training data, where the model's success rate is effectively zero (when the pass@$k$ rate remains zero even after sampling $k$ rollouts). Such cases are increasingly common in tasks requiring complex, multi-step reasoning (Yue et al., 2025; Wang et al., 2024). For these problems, an incorrect intermediate step can derail the entire reasoning chain for a 7B-scale LLM, resulting negative learning signals regardless of any partially correct solutions. Furthermore, naively penalizing all incorrect final outputs can further introduce training instability and hinder progress, making these difficult reasoning tasks largely intractable for standard outcome-based RL methods (Xiong et al., 2025).

An alternative approach is imitation learning, commonly implemented via Supervised Fine-Tuning (SFT) on expert demonstrations (Ross et al., 2011). While SFT can instill valuable reasoning behaviors, its next-token prediction objective enforces rigid, token-level imitation, limiting the model's ability to generalize beyond the training data. This problem becomes particularly pronounced when training data are modest in scale and when the model itself is relatively less capable. Under such conditions, long and complex demonstrations often lead to overfitting and shallow reasoning behaviors (Chu et al., 2025a; Li et al., 2025b), as illustrated by the performance decline in our Figure 1. Consequently, both SFT and outcome-based RL struggle on challenging reasoning tasks, leaving a critical gap for training small open-source models to effectively learn difficult problems.

To address this gap, we introduce Supervised Reinforcement Learning (SRL), a framework that reformulates problem-solving as a sequential decision-making process. Rather than optimizing for a final answer or imitating an entire expert trajectory, SRL trains the model to reproduce the sequence of key actions underlying expert reasoning, following an RL-style objective. Specifically, expert demonstrations are decomposed into a series of intermediate actions, each representing a meaningful decision step. During training, the model first generates an internal monologue to articulate its reasoning and then commits to an "action". At every step, SRL provides a reward based on the similarity between the model's predicted action and the corresponding expert action, thereby providing fine-grained, efficiently computable supervision that scales to large datasets.

Our work makes the following contributions:

- We propose SRL, a novel framework designed to enable effective learning on difficult reasoning tasks where SFT and RLVR struggle, by providing dense and smooth rewards based on similarity with expert actions.
- We demonstrate the effectiveness and robustness of SRL through extensive experiments on challenging mathematical reasoning and agentic software engineering benchmarks. Our results show that SRL significantly outperforms strong baselines across both domains (5.1 & 5.3).
- Through detailed analysis, we show that granular guidance is vital to SRL's reward and its impact on model behavior. We observe that SRL induces flexible and sophisticated reasoning patterns, such as interleaved planning and verification, which improve solution quality without simply increasing output length (5.2).

## 2 RELATED WORK

### 2.1 SFT (DISTILLATION) FOR LLM REASONING.

Distilling reasoning into smaller models via SFT on teacher-generated long Chain-of-Thought (CoT) rationales has proven highly effective for transferring complex problem-solving skills (Li et al., 2023; Huang et al., 2024; Min et al., 2024; Yeo et al., 2025), as exemplified by the small models distilled from DeepSeek R1 (Guo et al., 2025). Research indicates this process is surprisingly data-efficient, with small, high-quality datasets often being sufficient (Ye et al., 2025; Muennighoff et al., 2025).

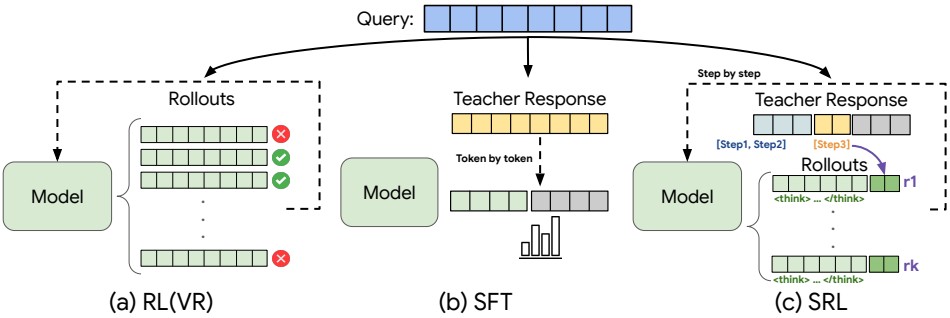

Figure 2: Illustration of SRL as compared to RL(VR) and SFT. **(a) RL(VR)** takes a query as input and performs k rollouts. The final answer correctness is used as the reward. **(b) SFT** uses both a query $\mathbf{x}$ and a complete teacher response $\mathbf{y}$ as input, training with a per-token loss to maximize the probability $p(\mathbf{y}|\mathbf{x})$. **(c) SRL** also uses a query and a teacher response. It breaks the response into step actions and, at each step, uses the previous steps as context. The model generates a next step action along with its step-wise inner thoughts, and the reward $r_k$ is based on the similarity between the model's and the teacher's action.

Given the success, research has focused on the underlying factor for effective SFT distillation (Chen et al., 2025a). Some emphasized the logical structure of the reasoning trace rather than its semantic correctness (Luo et al., 2025; Stechly et al., 2025), as models can learn from demonstrations with factual errors (Li et al., 2025a). Moreover, significant challenges remain in the student-teacher gap where the student fails to learn from overly complex data (Li et al., 2025b), and the risk of teacher hacking, where the student overfits to a teacher's specific flaws (Tiapkin et al., 2025). Ultimately, distillation from a teacher model imposes a performance ceiling on the student (Huang et al., 2024).

## 2.2 RL FOR LLM REASONING.

The development of DeepSeek-R1 (Guo et al., 2025) showed the effectiveness of rule-based RL for enhancing the reasoning capabilities of LLMs. This approach utilizes a scalable reward system based on final answer correctness, exemplified by the Group Relative Policy Optimization (GRPO) algorithm (Shao et al., 2024) and parallel algorithms (Ahmadian et al., 2024; Lambert et al., 2024; Xie et al., 2025). Building on this foundation, subsequent research has introduced numerous algorithmic refinements. For example, Dr. GRPO (Liu et al., 2025) mitigates bias by removing variance normalization, while DAPO (Yu et al., 2025) introduces a token-level loss and relaxes the policy update constraint by increasing the clipping threshold. Other notable advancements include modifications to clipping methods, normalization techniques, the KL divergence loss, and dynamic sampling strategies (Chu et al., 2025b; Zhang & Zuo, 2025; Zhang et al., 2025; Chen et al., 2025b). Despite these algorithmic variations, these approaches primarily rely on the final outcome's reward signal. A critical challenge arises when the rollouts fail to identify a correct solution trajectory, particularly for difficult queries. DAPO (Yu et al., 2025), for instance, addresses this by filtering out instructions that do not yield any successful rollouts.

## 3 PRELIMINARIES

A Large Language Model (LLM) is formally defined by a probability distribution $p_\theta$ over sequences of tokens, parameterized by a set of model weights $\theta$. Given an input prompt, represented as a token sequence $\mathbf{x} = [x_1, \ldots, x_n]$, the model generates a response sequence $\mathbf{y} = [y_1, \ldots, y_m]$. The response is produced autoregressively, where the generation of the token $y_j$ at any step $j$ is conditioned on the initial prompt $\mathbf{x}$ and all preceding tokens in the generated sequence, $(y_1, \ldots, y_{j-1})$. The joint probability of the entire response sequence $\mathbf{y}$ given the prompt $\mathbf{x}$ is thus factorized as: $p_\theta(\mathbf{y}|\mathbf{x}) = \prod_{j=1}^{m} p_\theta(y_j|\mathbf{x}, y_1, \ldots, y_{j-1})$.

**Supervised Fine-Tuning (SFT).** SFT is typically employed to specialize LLM for downstream applications or domains. It is also commonly used to establish a cold start for subsequent RL training phases that requires certain reply format/pattern, such as RL for reasoning (Deng et al., 2025) or tool

use (Feng et al., 2025). Specifically, the process utilizes a dataset $\mathcal{D} = \{(\mathbf{x}^{(i)}, \mathbf{y}^{(i)})\}_{i=1}^{N}$, where each $\mathbf{x}^{(i)}$ is an input prompt and $\mathbf{y}^{(i)}$ is the corresponding desired model output. The primary objective is to update the parameters $\theta$ to maximize the conditional probability of generating the target response $\mathbf{y}^{(i)}$ given the input prompt $\mathbf{x}^{(i)}$. This goal is formally achieved by minimizing the negative log-likelihood loss function: $\mathcal{L}_{\text{SFT}}(\theta) = -\sum_{i=1}^{N} \log p_{\theta}(\mathbf{y}^{(i)}|\mathbf{x}^{(i)})$ over the entire dataset. By minimizing this loss, the model learns to produce responses that are closely aligned with the exact words demonstrated in the labeled training examples.

**Reinforcement Learning (RL).** Recent literature on improving model reasoning capability has focused on RL with verifiable reward (RLVR), where the policy model receives reward signals purely based on the final answer correctness. Building on this principle, Group Relative Policy Optimization (GRPO) (Shao et al., 2024) involves sampling a group of $G$ response trajectories, $\{\mathbf{o}_i\}_{i=1}^{G}$, from the previous policy model, $\theta_{old}$, for each input query $\mathbf{x}$. The objective function for GRPO is:

$$\mathbb{E}\left[\frac{1}{G}\sum_{i=1}^{G}\frac{1}{|\mathbf{o}_i|}\sum_{t=1}^{|\mathbf{o}_i|}\min\left(\frac{p_\theta(o_{i,t}\,|\,\mathbf{x},\mathbf{o}_{i,<t})}{p_{\theta_{old}}(o_{i,t}\,|\,\mathbf{x},\mathbf{o}_{i,<t})}\hat{A}_{i,t},\,\text{clip}\left(\frac{p_\theta(o_{i,t}\,|\,\mathbf{x},\mathbf{o}_{i,<t})}{p_{\theta_{old}}(o_{i,t}\,|\,\mathbf{x},\mathbf{o}_{i,<t})},1-\epsilon,1+\epsilon\right)\hat{A}_{i,t}\right)\right]$$
$$-\,\beta\,\mathbb{D}_{\text{KL}}\Big[p_\theta\,\|\,p_{\text{ref}}\Big]. \tag{1}$$

The hyperparameter $\epsilon > 0$ defines the clipping range for the policy update ratio, and the coefficient $\beta > 0$ modulates the influence of the KL-divergence penalty against the policy update. The term $\theta_{old}$ refers to the policy from the previous iteration. The advantage function, $\hat{A}_{i,t} = (\tilde{r}_i - \text{mean}(\tilde{r}))/\text{std}(\tilde{r})$, is defined as the group-level normalized reward.

A key challenge for these RL algorithms emerges when input queries are either too easy or too hard, resulting in uniform correctness within policy rollouts $\{\mathbf{o}_i\}_{i=1}^{G}$. In such cases, the advantage estimate $\hat{A}_{i,t}$ vanishes, yielding an uninformative policy gradient and preventing model updates. A common strategy to mitigate this is to dynamically sample the batches, filtering out samples and re-sampling until the data sample satisfies $0 < |\{\mathbf{o}_i|\text{is\_correct}(\mathbf{o}_i)\}| < G$ (Yu et al., 2025).

## 4 METHODOLOGY

### 4.1 THE CHALLENGE OF HARD REASONING PROBLEM

RL with verifiable reward is a prominent technique for enhancing the reasoning capabilities of LLMs. The strategy is to close the gap between a model's potential performance across multiple attempts (pass@$k$) (Brown et al., 2024; Yue et al., 2025). However, this paradigm falters on problems where the model's pass@$k$ rate is already near zero. For this set of difficult problems, which we term $\mathcal{D}_{\text{hard}}$, positive reward signals are too sparse for RLVR to be effective (Xiong et al., 2025). Moreover, simply penalizing incorrect outputs can be detrimental to model performance (Yu et al., 2025; Xiong et al., 2025), creating a significant challenge for improving model reasoning.

Formally, we define $\mathcal{D}_{\text{hard}} = \{\mathbf{x}^{(i)}, a^{(i)}\}_{i=1}^{N}$ as the set of problems $(\mathbf{x}, a)$ where policy model's success rate is low with $k$ samples: $\frac{1}{k}\sum_{j=1}^{k}\mathbb{I}\big(\text{ExtractAnswer}(\mathbf{y}^{(j)}) == a\big) \leq \epsilon$, where each solution attempt $\mathbf{y}^{(j)}$ is sampled from the policy $p_{\theta}(\cdot|\mathbf{x})$ and $\epsilon > 0$ is a small constant.

Due to the scarcity of successful trajectories, standard RL with verifiable reward struggles on $\mathcal{D}_{\text{hard}}$. Such data is further difficult to be learned by SFT, due to its limited amount and complexity in teacher reasoning trajectories (Li et al., 2025b).

### 4.2 SUPERVISED REINFORCEMENT LEARNING (SRL)

To address the challenge of learning from $\mathcal{D}_{\text{hard}}$, we introduce Supervised Reinforcement Learning (SRL), a framework that decomposes complex problem-solving as a sequential decision-making process, and thus can be easily learned on how to properly operate step-wise. Instead of generating a monolithic solution, the model learns to take actions similar to the expert while producing their own inner reasoning process in a step-by-step manner. The whole framework is illustrated in Figure 3. [1]

---

[1]Empirically, we found that providing the subsquent step title (e.g., "2. **Coprime Pairs**" in Figure 3) as the additional context for the learner to predict the rest of the step content can further boost the performance.

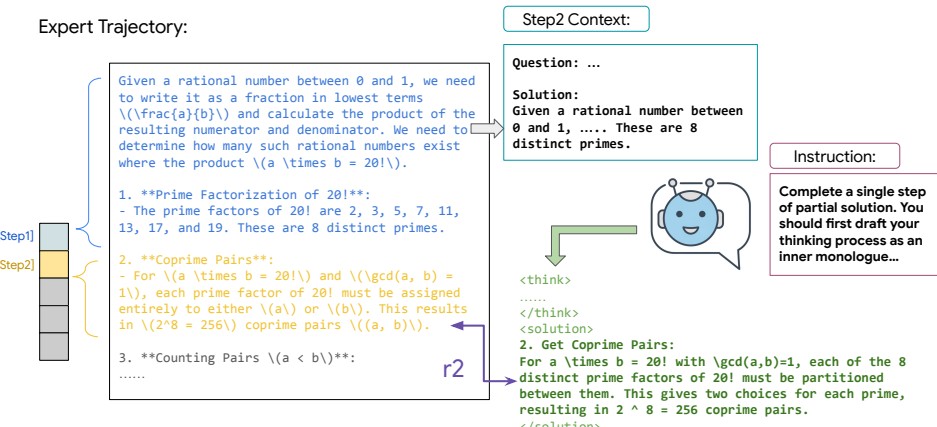

Figure 3: Given a solution trajectory, we take each summarized step as an action to be learned and take the partial solution before the step as the context of our newly created data. The model is then prompted to generate its thinking process followed by the action for the current step. A reward ($r_2$ in the figure) is then calculated based on the similarity between the model's and the expert's action.

**Action-based problem formulation.** Given an expert solution trajectory $\mathbf{y}$ that leads to a correct final answer, we decompose $\mathbf{y}$ into a sequence of tuples: $\mathbf{y} = \{\mathbf{y}_{\text{step}_n}\}_{n=1}^N$. Each step represents a *logical action*: the concrete action to be operated. This formulation is domain-agnostic; for instance, an action in mathematical reasoning could be an algebraic manipulation, while for a software agent, it could be a command executed in a code repository.

**Step-wise training data construction.** To create training data for SRL, we leverage a powerful teacher model, $\boldsymbol{\theta}_{expert}$, to generate solution trajectories. From a single complete solution with $N$ steps, we construct $N - 1$ partial trajectories. For each step $k \in \{1, \ldots, N-1\}$, we create a new input prompt $\mathbf{x}_{\text{step}_k} = [\mathbf{x}, \mathbf{y}_{\text{step}_1}, \ldots, \mathbf{y}_{\text{step}_{k-1}}]$, where the model's task is to predict the subsequent step, $\mathbf{y}_{\text{step}_k}$. This process transforms one expert solution into a rich set of training instances that teach the model to proceed correctly from various intermediate states.

**Learning with a sequence similarity reward with own inner monologue.** Given a partial context $\mathbf{x}_{\text{step}_k}$ containing the problem and a partial solution, the policy model $p_{\boldsymbol{\theta}}$ is prompted to generate the subsequent action step with their own inner monologue $\mathbf{y}'_{\text{think}}$, which is encapsulated by "<think>" tags. We then provide a dense reward based on the quality of the generated logical action $\mathbf{y}'_{\text{step}_k}$. The prediction can be formally specified as: $\mathbf{y}' \sim p_{\boldsymbol{\theta}}(\cdot | \mathbf{x}_{\text{step}_k}) = [\mathbf{y}'_{\text{think}}, \mathbf{y}'_{\text{step}_k}]$.

To guide training, we consider the reward function that measures the similarity between the generated action: $R(\mathbf{y}'_{\text{step}_k}, \mathbf{y}_{\text{step}_k}) = \frac{2M}{T}$, where

- **T (Total elements):** This is the total number of elements in both sequences combined. It is calculated as the sum of the lengths of the two sequences: $T = |S_1| + |S_2|$.
- **M (Matched elements):** The total count of elements found in all non-overlapping matching blocks between the two sequences. The algorithm first finds the longest contiguous matching subsequence and then recursively searches for more matches in the segments to the left and right of that block. If we represent the set of all such matching blocks as a list of tuples $(i, j, n)$, where $n$ is the length of the matching block, then $M$ is the sum of all lengths $n$: $M = \sum_{(i,j,n) \in \text{MatchingBlocks}} n$.

Combining these definitions, we can calculate the similarity ratio $R \in [0, 1]$ as:

$$R = \frac{2 \sum_{(i,j,n) \in \text{MatchingBlocks}} n}{|S_1| + |S_2|}$$

While the choice of similarity function can be adapted to specific data distributions or task requirements, we employ Python's `difflib.SequenceMatcher` for our implementation. Furthermore, a negative reward is assigned if the generated output $\mathbf{y}'$ fails to adhere to the required format. Hence,

the final reward used is:

$$r(\mathbf{y}'_{\text{step}_k}, \mathbf{y}_{\text{step}_k}) = \begin{cases} R(\mathbf{y}'_{\text{step}_k}, \mathbf{y}_{\text{step}_k}) & \text{if } \mathbf{y}' \text{ follows format,} \\ -1 & \text{otherwise.} \end{cases}$$

The policy $p_{\boldsymbol{\theta}}$ is then optimized using this reward signal with the GRPO objective function defined in Equation 1. Notably, our reward is computed only on the logical action, not the internal monologue. This grants the model flexibility to develop its own internal reasoning style while ensuring its external actions align with the expert's strategy. This design provides dense, step-level feedback and allows for rapid reward calculation, making the SRL framework both effective and scalable.

**Dynamic sampling for SRL.** As our reward signal $r \in [0,1] \cup \{-1\}$ is dense, we generalize the dynamic sampling strategy previously designed for outcome accuracy and implement it to filter samples with less meaningful updates. Specifically, a sample should be filtered out if its rollouts yield rewards with near-zero variance, providing a weak advantage and thus weak learning signal. With the sequence similarity reward in SRL, we retain a sample if the standard deviation of the reward scores of its rollouts exceeds a threshold $\epsilon > 0$:

$$\sqrt{\frac{\sum_{i=1}^{G}(r(\mathbf{o}_i, \mathbf{y}) - \bar{r})^2}{G}} > \epsilon$$

where $G$ is the number of generated rollouts, $r(\mathbf{o}_i, \mathbf{y})$ is the sequence similarity reward for the $i$-th rollout $\mathbf{o}_i$ given the expert trajectory $\mathbf{y}$, and $\bar{r}$ is the mean reward for the sample. To maintain a consistent batch size of $B$, we continuously sample and filter until the batch is filled.

# 5 EXPERIMENTS

## 5.1 MAIN RESULTS: MATH REASONING

**Setup.** We finetue *Qwen2.5-7B-Instruct* (Yang et al., 2024) on the *s1K-1.1* dataset (Muennighoff et al., 2025). This dataset contains 1,000 diverse and challenging questions, each accompanied by a detailed reasoning trace and a final solution generated by DeepSeek R1. The solutions from DeepSeek R1 are formatted with structured, numbered steps (e.g., "1. Title of Step 1"). We leverage this structure to generate intermediate training targets by parsing these solutions and treating each complete step as a ground-truth continuation. Any data points that do not adhere to this format are excluded. The detailed data preparation process can be found in Appendix B.3. We hold out 60 questions from the dataset to form our validation set.

**Baselines.** We benchmark our proposed methods against several baselines, all initialized from the *Qwen2.5-7B-Instruct* model. These baselines include:

1. **SFT** on either the complete reasoning traces (R1 reasoning) or the final solutions from the s1K-1.1 dataset (R1 outline)
2. **s1K-7B**, the official distilled model released by the dataset's authors
3. **RLVR**, which we implement using the GRPO algorithm. To ensure fair comparison, we implement additional dynamic sampling as in Yu et al. (2025), which removes samples with all correct or incorrect rollouts.
4. **SFT → RLVR**, the same RLVR algorithm applied after an initial SFT phase
5. **R³** (Xi et al., 2024), which applies a reverse curriculum RL learning strategy by starting near the end of the demonstration solution and working backward with step-by-step decomposition to make the sparse final reward easier to find [2].

Our proposed method, SRL, is likewise evaluated both as a standalone technique and in a sequential configuration where it precedes RLVR (SRL then RLVR). All models are trained for up to 30 epochs, and for each method, we select the checkpoint with the best performance on the validation set.

**Evaluation.** We evaluate all models on the following four competition-level mathematical reasoning benchmarks: AMC23, AIME24, AIME25[3] and Minerva Math (Lewkowycz et al., 2022). Our

---

[2] We re-implement this method by reformulating the data from *s1K-1.1*.

[3] Data available on HuggingFace (https://huggingface.co/datasets): `AI-MO/aimo-validation-amc`, `AI-MO/aimo-validation-aime`, and `math-ai/aime25`

evaluation protocol for all benchmarks follows the package released by Qwen2.5-Math[4] and report the accuracy of greedy sampling. The evaluation prompt can be viewed in Appendix B.2.2. In addition, for AMC23, AIME24 and AIME25, we report the average@32 score with a temperature of 1.0 for all baselines to ensure a more robust evaluation, referenceing the evaluation in Yu et al. (2025).

**Performance.** The performance results of our models are summarized in Table 1. Consistent with the officially released S1K-7B model, our model trained with SFT on the same dataset exhibited a notable performance degradation. In contrast, methods based on RL maintained generalization on the evaluation benchmarks. Specifically, while RLVR maintained the performance, SRL provided a substantial boost of 3.0% on average. Furthermore, applying RLVR after SRL training yielded a 3.7% increase on average, leveraging only 1k training data.

Comparatively, while the baseline $R^3$ also use teacher demonstrations to structured a step-by-step learning, it exhibits distinct performance characteristics. As shown in Table 1, $R^3$ performs competitively on AMC23 but lags behind SRL on the more challenging AIME24 and AIME25 benchmarks (e.g., SRL achieves 16.7% vs. $R^3$'s 13.3% on AIME24). This divergence highlights a critical difference between SRL and $R^3$'s objective, which relies on a reverse curriculum to facilitate exploration towards a sparse outcome reward. In contrast, SRL employs dense, intrinsic supervision, rewarding the model for the **quality** of its step-wise "actions" rather than just the final answer. Consequently, for complex reasoning tasks where outcome-based exploration is very challenging, SRL provides a more robust learning signal, effectively teaching the model *how* to reason and act.

Table 1: Evaluation results across competition-level math benchmarks. We take Qwen2.5-7B-Instruct as the base model and report the performance of different training schemes (SFT, RLVR via GRPO, and SRL) using the same set of training data. The **bold** numbers indicate the best results among the open-source models and the underscored numbers represent the second-best results.

| Model | AMC23 | | AIME24 | | AIME25 | | Minerva Math | Average |
|---|---|---|---|---|---|---|---|---|
| | Avg@32 | Greedy | Avg@32 | Greedy | Avg@32 | Greedy | | |
| *Base Model* | | | | | | | | |
| Qwen2.5-7B-Instruct | 49.3 | 50.0 | 10.5 | 13.3 | 7.5 | 6.7 | 34.9 | 24.6 |
| *Training with SFT* | | | | | | | | |
| S1K-7B | 24.1 | 25.0 | 2.2 | 3.3 | 3.7 | 3.3 | 20.2 | 11.7 |
| SFT (R1 reasoning) | 26.8 | 40.0 | 3.9 | 10.0 | 5.4 | 10.0 | 20.2 | 16.6 |
| SFT (R1 outline) | 36.2 | 27.5 | 5.1 | 3.3 | 3.8 | 6.7 | 31.6 | 16.3 |
| *Training with Outcome-Reward RL* | | | | | | | | |
| RL(VR) | 52.0 | 47.5 | 11.1 | 10.0 | 7.4 | 10.0 | 33.8 | 24.5 |
| SFT (outline) → RL(VR) | 37.6 | 35.0 | 4.9 | 3.3 | 4.5 | 6.7 | 30.1 | 17.4 |
| $R^3$ | **52.5** | **57.5** | 10.0 | 13.3 | 7.4 | 10.0 | 35.3 | 26.6 |
| *Training with SRL* | | | | | | | | |
| SRL | 51.5 | 50.0 | 13.2 | 16.7 | 7.1 | **13.3** | **36.4** | 27.6 |
| SRL → RLVR | 52.1 | **57.5** | **13.3** | **20.0** | **8.6** | 10.0 | **36.4** | **28.3** |

## 5.2 ANALYSIS: MATH REASONING

**Effect of dynamic sampling in SRL.** In Table 2, we analyze the impact of the dynamic sampling component in SRL, based on thresholding the standard deviation of sequence similarity rewards within rollouts. For both models, we train until the training reward converges and select checkpoint based on validation scores. Our results are consistent the findings of DAPO (Yu et al., 2025), which stated that removing samples that provide a zero learning signal is critical in the effectiveness of the RL training loop, showing non-trivial d performance improvement.

**Disentangling the impact of guidance granularity in SRL.** To isolate the effects of guidance granularity from the benefits of sequence similarity rewards, we evaluated our multi-step method against two simplified, one-step baselines:

- Final-answer reward: This baseline utilizes RLVR trained with GRPO, rewarding the model solely based on the correctness of the final answer.

---

[4]https://github.com/QwenLM/Qwen2.5-Math

Table 2: The effect of dynamic filtering on SRL. Filtering out samples with less meaningful updates provides non-trivial performance improvement. DS stands for dynamic sampling.

| Model | AMC23 | | AIME24 | | AIME25 | | Minerva Math | Average |
|---|---|---|---|---|---|---|---|---|
| | Avg@32 | Greedy | Avg@32 | Greedy | Avg@32 | Greedy | | |
| SRL w/out DS | 48.5 | **52.5** | 11.1 | 13.3 | 6.8 | 6.7 | 33.8 | 24.7 |
| SRL w/ DS | **51.5** | 50.0 | **13.2** | **16.7** | **7.1** | **13.3** | **36.4** | **27.6** |

Table 3: Model Performance of different reward functions and density. For sequence similarity reward, we implement it with the entire expert output as a one-step supervision. The model benefits from our multi-step decomposition on the small set of challenging training data.

| Model | AMC23 | | AIME24 | | AIME25 | | Minerva Math | Average |
|---|---|---|---|---|---|---|---|---|
| | Avg@32 | Greedy | Avg@32 | Greedy | Avg@32 | Greedy | | |
| *Final Answer Reward* | | | | | | | | |
| One-step (RLVR) | **52.0** | 47.5 | 11.1 | 10.0 | **7.4** | 10.0 | 33.8 | 24.5 |
| *Sequence Similarity Reward* | | | | | | | | |
| One-step | 51.8 | **52.5** | 11.7 | 13.3 | 6.9 | 10.0 | 34.9 | 25.9 |
| Multi-step (SRL) | 51.5 | 50.0 | **13.2** | **16.7** | 7.1 | **13.3** | **36.4** | **27.6** |

- Holistic sequence similarity reward: The model generates a complete solution in a single step. This entire solution is then evaluated for similarity against the complete ground-truth trajectory.

As shown in Table 3, the results underscore the value of fine-grained guidance for sequence similarity reward. While the holistic sequence-similarity reward improved performance on average, providing granular, step-by-step guidance led to markedly superior performance across benchmarks.

**Model size variation.** To evaluate the scalability of our approach, we apply our method to *Qwen2.5-3B-Instruct* (Yang et al., 2024) using the identical data setup from Table 1. As summarized in Table 4, our method yields a consistent performance boost, improving the average score over base model by 3.1% and surpassing RL(VR). Notably, while the limited capacity of the smaller base model restricts improvements on extremely difficult benchmarks (e.g., AIME24 & AIME25), we observe significant gains on more tractable tasks like AMC23 and Minerva Math.

Table 4: Evaluation results across competition-level math benchmarks using smaller base model. We take Qwen2.5-3B-Instruct as the base model in this table to test the generalization of our method.

| Model | AMC23 | | AIME24 | | AIME25 | | Minerva Math | Average |
|---|---|---|---|---|---|---|---|---|
| | Avg@32 | Greedy | Avg@32 | Greedy | Avg@32 | Greedy | | |
| Qwen2.5-3B-Instruct | 32.5 | 45.0 | 3.3 | 3.3 | 1.7 | 3.3 | 25.7 | 16.4 |
| RL(VR) | 35.0 | 42.5 | 6.7 | 3.3 | 6.7 | 3.3 | 26.5 | 17.7 |
| SRL → RLVR | 37.5 | 47.5 | 6.7 | 3.3 | 6.7 | 6.7 | 28.3 | 19.5 |

**Interleaved reasoning behaviors.** Beyond achieving superior performance, models trained with SRL exhibit distinctive and adaptive reasoning trajectories. While conventional models typically generate a single, monolithic reasoning block at the outset, our SRL-finetuned models—particularly those trained with RLVR—dynamically interleave reasoning steps throughout the generation process. This behavior manifests in three primary modes: (1) **Structured Planning**, where the model outlines a comprehensive roadmap before execution; (2) **Dynamic Adjustment**, where the model inserts iterative reasoning blocks to refine its trajectory mid-solution; and (3) **Reflective Verification**, where the model pauses to evaluate its logic and verify the answer before committing to a final output. These emergent patterns are illustrated in Example 1 and Example 2 (see Appendix), which showcase the model's transition from static generation to an integrated, "think-and-act" reasoning process. For further qualitative analysis and studies, please refer to Appendix C.

---

**Example 1: Interleaved thinking and self-verification**

Question: What is the least positive integer value of $x$ for which the inequality $3x > 2x + 1$ is true?

$<$think$>$
To solve the inequality $3x > 2x + 1$, we can start by isolating $x$. Let's subtract $2x$ from both sides of the inequality.
$</$think$>$

Subtract $2x$ from both sides:
$$3x - 2x > 2x + 1 - 2x$$

This simplifies to:
$$x > 1$$

$<$think$>$
We can verify this by substituting $x = 2$ into the original inequality to ensure it holds true.
$</$think$>$

The inequality $x > 1$ tells us that $x$ must be greater than 1. The smallest positive integer greater than 1 is 2. Therefore, the least positive integer value of $x$ that satisfies the inequality is 2.

Substitute $x = 2$ into the original inequality:
$$3(2) > 2(2) + 1$$

This simplifies to:
$$6 > 5$$

This is true, so $x = 2$ is indeed a solution.

Therefore, the least positive integer value of $x$ for which the inequality $3x > 2x + 1$ is true is $\boxed{2}$.

---

## 5.3 EXTENSION: SOFTWARE ENGINEERING AGENTIC REASONING

**Task.** We extend our SRL framework to the domain of software engineering, training agents to resolve real-world programming issues. These tasks are commonly evaluated on benchmarks like SWE-Bench (Jimenez et al., 2023), which requires agents to perform complex, multi-turn interactions with large codebases and reason deeply about code functionality.

However, unlike in math domain, the direct application of online RL to software engineering is impeded by significant practical challenges. These include handling long context windows, high-latency environment feedback, and slow patch verification (Wei et al., 2025; Golubev et al., 2025). Consequently, these obstacles have hindered the development of stable and scalable end-to-end RL methods, leading to a prevailing approach of collecting expert agent trajectories and distilling them into a policy via SFT (Pan et al., 2024; Yang et al., 2025).

**Setup.** We apply SRL to further fine-tune *Qwen2.5-Coder-7B-Instruct* (Hui et al., 2024), a model already specialized for coding tasks. We use a dataset from Yang et al. (2025), which consists of 5,000 expert agent trajectories. These trajectories were generated by *claude-3-7-sonnet-20250219* (Anthropic, 2025) and subsequently verified to ensure they produce correct code patches.

Each trajectory is composed of multiple steps defined by the agent's interactions with the coding environment. As the example below illustrates, a single step contains natural language reasoning followed by an executable action:

```
I'll help you implement the necessary changes to fix the issue with the `OriginValidator` not
properly handling wildcard (*) in allowed_origins. Let's follow the steps you outlined.

## Step 1: Find and read relevant code

First, let's explore the repository structure to locate the `OriginValidator` class mentioned
in the PR description. This is an extra long line added to demonstrate how the automatic line
wrapping feature from the listings package works.

<function=bash>
<parameter=command>find /testbed -type f -name "*.py" | grep -v "__pycache__" | sort</paramete
r>
</function>
```

In line with our SRL formulation (Section 4.2), we define the "action" as the environment-consumable command (e.g., the bash call). Following this decomposition, we process the full trajectories to create

134k step-wise training instances. For validation, we hold out 30 full trajectories, from which we curate a validation set of 650 step-wise instances.

**Evaluation.** We evaluate our model's patch generation performance by measuring its resolve rate (%) under two distinct configurations, following Wei et al. (2025): (1) Oracle file editing evaluation: The model is provided with the oracle code files to repair. This configuration isolates and measures the model's core patch generation capability; (2) End-to-end evaluation: This setting uses the Agentless-mini agent scaffold (Wei et al., 2025) to first identify the file(s) to modify and subsequently generate the patch. It tests the model's fault localization and code repair abilities in conjunction.

We compare our SRL-trained model against two crucial baselines: the original base model (*Qwen2.5-Coder-Instruct*) and SWE-Gym-7B (Pan et al., 2024). Since SWE-Gym-7B is an SFT-based model finetuned from the same base model, this provides a direct, fair comparison between SFT and our SRL training methodology. As shown in Table 5, SRL substantially outperforms both baselines. In the oracle setting, SRL achieves a 14.8% resolve rate, representing a 74% relative improvement over the strong SWE-Gym-7B baseline. The performance gain is consistent when evaluating in the challenging end-to-end setting, where SRL can obtain twice the performance.

Table 5: Performance of SRL on SWE-Bench-Verified. Results in the table are using greedy decoding.

|  | Oracle File Edit | End-to-End |
| --- | --- | --- |
| Qwen2.5-Coder-Instruct (Base) | 5.8 | 3.2 |
| SWE-Gym-7B | 8.4 | 4.2 |
| SRL (ours) | **14.8** | **8.6** |

## 5.4 DISCUSSION AND LIMITATIONS

Despite the strong performance of SRL, its effectiveness is subject to several inherent constraints. Primarily, the method relies heavily on the availability of high-quality, decomposable expert trajectories, as it assumes a well-defined "action" granularity where complex reasoning can be cleanly partitioned into discrete steps. This dependency suggests that SRL may be less effective in domains where expert demonstrations are sparse or in tasks that require holistic intuition rather than a sequential chain-of-thought structure. Furthermore, the success of the self-reflective loop is fundamentally tied to the student model's initial proficiency; the model must already possess a baseline instruction-following capability to generate rollout samples that are coherent enough to yield non-trivial reward signals. Without this initial competence, the learning process may suffer from excessively sparse rewards. Consequently, future research will focus on adapting SRL to domains with ambiguous action definitions, exploring techniques to alleviate the dependency on highly structured expert data, and investigating the variance in performance when applying different similarity functions to guide the learning process.

## 6 CONCLUSION

In conclusion, we introduced Supervised Reinforcement Learning (SRL), a novel method designed to teach LLMs complex reasoning skills from expert demonstrations, particularly for problems that are too difficult for conventional RL or SFT approaches. By breaking down expert solutions into manageable steps and leveraging a dense sequence similarity reward, SRL provides effective, granular guidance that bridges the gap between imitation learning and reinforcement learning. Our empirical results demonstrate that SRL not only significantly outperforms baseline methods in both mathematical reasoning and software engineering tasks but also enables a powerful curriculum learning strategy when combined with RLVR. This work establishes SRL as a robust and generalizable technique for unlocking a model's potential to learn from challenging, multi-step problems, paving the way for training more capable and versatile AI agents.

## 7 ACKNOWLEDGMENTS

We thank Jinwei Xing, Jinsung Yoon, and members from Google Cloud AI Research for their valuable feedback during the preparation of the paper.

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

## A    ILLUSTRATION OF SRL ON SWE TASKS.

In Figure 4, we illustrate how we approach the SWE tasks with SRL. We take two consecutive action-observation pairs from the expert trajectories in the given SFT data as context. We prompt the LLM to first think in monologues and then conclude with its action. Sequence similarity score is thus computed between model action and expert action in the trajectory.

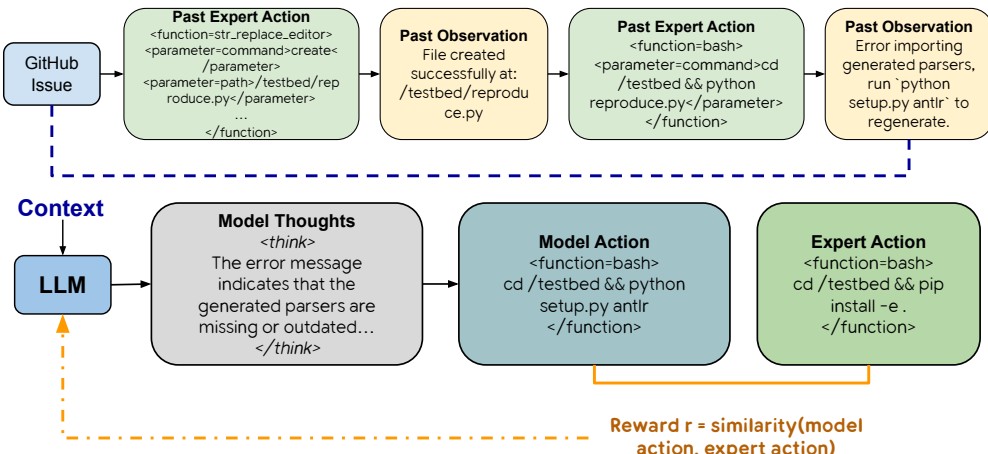

Figure 4: Illustration of applying SRL to SWE tasks. We take two rounds of the past expert actions and corresponding observations as context and prompt the LLM to think before reaches the next action. Reward is then calculated via comparing model action with the expert action.

## B    EXPERIMENT DETAILS

### B.1    COMPUTATION AND PARAMETER DETAILS

We use VeRL as the training framework for GRPO and implement SRL based on it. In Table 6, 7 and 8, we detail the hyperparameters that we used for SFT, GRPO and SRL. The batch size for GRPO is set to be smaller (128) than SRL (512) as the filter rate is high based on final answer correctness, for most of the questions are very difficult. We maintain a smaller batch size to avoid repeated sampling and achieve better performance. We further lay out the system prompts we used for SRL. Experiments were conducted on GPU clusters to the similar level of NVIDIA H100 80GB GPU.

Table 6: Supervised fine-tuning hyperparameters.

| | |
|---|---|
| Data type | bf16 |
| Learning rate | 5e-6 |
| Global batch size | 64 |
| Scheduler | Cosine |
| Warmup ratio | 0.3 |
| Num train epochs | 3 |

Table 7: GRPO hyperparameters.

| | |
|---|---|
| Batch size | 128 |
| Max grad norm | 1.0 |
| Data type | bf16 |
| Learning rate | 5e-7 |
| Warmup ratio | 0.0 |
| Rollout temperature | 1.0 |
| Rollout number | 8 |
| KL loss coeff | 0 |

Table 8: SRL hyperparameters.

| | |
|---|---|
| Batch size | 512 |
| Max grad norm | 1.0 |
| Data type | bf16 |
| Learning rate | 5e-7 |
| Warmup ratio | 0.0 |
| Rollout temperature | 1.0 |
| Rollout number | 8 |
| KL loss coeff | 0 |

## B.2 PROMPT DETAILS

### B.2.1 SYSTEM PROMPT FOR SRL LEARNING

```
System Prompt for SRL

You are a helpful assistant for solving mathematical problems. A user will provide a math problem, which
    may include a partial solution. Your task is to continue the solution by providing the very next
    logical step.

A user will ask you to solve a task. You should first draft your thinking process (inner monologue).
    Then, generate the solution.

Your response format must follow the template below:
<think> Your thoughts or/and draft, like working through an exercise on scratch paper. Be as casual and
    as long as you want until you are confident to generate a correct solution. </think>

Provide only the single, next step to continue the solution. Do not solve the entire problem.
```

### B.2.2 SYSTEM PROMPT FOR EVALUATION

```
System Prompt for Math Evaluation

Please reason step by step, and put your final answer within \boxed{}. You can draft your thinking
    process inside <think> </think> tags anywhere. This is your "scratch paper".
```

## B.3 TASK DECOMPOSITION DETAILS

We conduct the task decomposition for math via a data re-writing process. We first decide a format that is easy to parse trajectory into steps:

```
Template Format for Task Decomposition in Math

SOLUTION:

1. **{{STEP TITLE}}**
{{STEP ACTION}}

2. **{{STEP TITLE}}**
{{STEP ACTION}}

3. ....
```

Then we prompt Gemini-2.5-flash (Team, 2025) to conduct data conversion using the below prompt:

---

**Prompt Used for Data Rewrite in Math**

```
### ROLE
You are an expert Math Tutor and Educational Content Formatter. Your goal is to take a raw math problem
    and its solution, and rewrite it into a strictly structured, easy-to-follow, step-by-step guide.

### INPUT DATA
I will provide you with:
1. The Math Question.
2. The Original Solution.

### OUTPUT FORMATTING RULES
You must adhere to the following format strictly:

```
SOLUTION:

1. **{{STEP TITLE}}**
{{STEP ACTION}}
2. **{{STEP TITLE}}**
{{STEP ACTION}}
3. ... [Continue until solved]
```

### CONTENT GUIDELINES
1. **Step Titles:** Must be bolded (**Title**). They should be short, imperative verb phrases (e.g., "
    Identify the variables," "Isolate x," "Substitute values").
2. **Step Actions:** The equations and operations to execute according to the step title
3. Please follow the original thoughts and solutions provided in the ``Original Solution''. Do not solve
    the task by your own.

### EXAMPLE
[[TWO SHOT EXAMPLE PUT HERE]]

### CURRENT TASK
**Question:** [INSERT QUESTION HERE]
**Original Solution:** [INSERT SOLUTION HERE]
```

For the data after conversion cannot pass our formatting check, we will discard them.

## C  SUPPLEMENTARY STUDIES

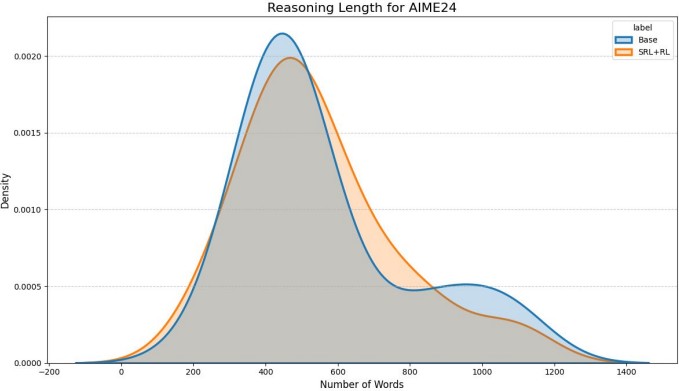

Figure 5: Reasoning length distribution for base model and model trained with SRL.

In this section, we provide further analysis to supplement our experimental results in Section 5.2. Specifically, we aim to illustrate the mechanisms behind the performance improvements of our SRL-finetuned model. While recent work often associates improved reasoning with longer generation sequences, we first demonstrate that our model's gains are not a result of increased verbosity. In addition, we show that the model develops a more sophisticated internal structure. Through the following examples of **structured planning** and **post-hoc self-verification**, we visualize how the model leverages specific reasoning patterns to navigate complex problem-solving tasks more effectively than the base model.

**Reasoning length.** We investigate whether the performance gains from SRL can be simply attributed to an increase in reasoning length. In Figure 5, we plot the distribution of reasoning lengths (i.e., the number of words in the solution) for both the base model and our SRL-finetuned model. We observe no significant difference between the two distributions. This finding suggests that the performance improvement stems from enhanced planning and higher-quality reasoning, rather than merely an increased token budget at inference time.

**Qualitative Case Studies.** The first example demonstrates *a priori* structured planning, while the second showcases the model's ability to perform mid-stream or post-hoc self-verification.

---

**Example 2: Planning**

Question: Josh decides to try flipping a house. He buys a house for $80,000 and then puts in $50,000 in repairs. This increased the value of the house by 150%. How much profit did he make?
<think>
To determine Josh's profit, we need to follow these steps:
1. Calculate the new value of the house after the repairs.
2. Subtract the initial cost of the house and the repair costs from the new value to find the profit.
</think>
First, let's calculate the new value of the house after the repairs. The value of the house increased by 150% after the repairs.
The initial cost of the house is $80,000, and the repairs cost $50,000. The total cost before the value increase is:

$$80,000 + 50,000 = 130,000$$

The value of the house increased by 150% of its initial value of $80,000. So, the increase in value is:

$$150\% \times 80,000 = 1.5 \times 80,000 = 120,000$$

Therefore, the new value of the house is:

$$80,000 + 120,000 = 200,000$$

Now, we subtract the total cost (initial cost of the house plus the repair costs) from the new value of the house to find the profit:

$$200,000 - 130,000 = 70,000$$

So, Josh's profit is:

70000

---

**Example 3: Self-verification**

Question: Which one of these constellations is not located along the Milky Way in the sky?
Answer Choices: (A) Perseus (B) Cygnus (C) Scorpius (D) Leo

First, let's recall which constellations are along the Milky Way. Perseus, Cygnus, and Scorpius are all well-known constellations that lie along the Milky Way. However, Leo is not one of them; it is located in the northern sky and does not align with the Milky Way. Therefore, the correct answer is (D) Leo.

<think>
To double-check, I can recall the general locations of these constellations:
- Perseus is in the northern sky and is part of the Perseus Arm of the Milky Way.
- Cygnus is also in the northern sky and is part of the Orion Arm of the Milky Way.
- Scorpius is in the southern sky and is part of the Sagittarius Arm of the Milky Way.
- Leo is in the northern sky and is not aligned with the Milky Way.
</think>

The answer is (D) Leo.

---

# D    LLM USAGE

Throughout the preparation of this manuscript, we utilized a LLM as an assistant with refinement of writing. Its role was strictly limited to improving grammar, rephrasing sentences for clarity, and

correcting spelling. All intellectual contributions, ideas, and conclusions presented in this paper are entirely the work of the authors.

