# OpenReview forum: "Supervised Reinforcement Learning: From Expert Trajectories to Step-wise Reasoning"
_ICLR.cc/2026/Conference — ICLR 2026 Poster_

### Official Review · Reviewer_iZLY · 2025-10-27

**Soundness:** 3
**Presentation:** 3
**Contribution:** 3
**Rating:** 6
**Confidence:** 2

**Summary:**

The authors present a method for supervised or assisted RL for LLMs.
The motivation given for the approach is that for difficult tasks, where the LLM rarely reaches the right answer, naive use of RLVR does not help.
Thus, the authors propose to break the process of generating an answer using the LLM into multiple steps.
At each step the LLM is given the problem and a partial answer, and is required to output the only the next step (action) to solve the problem after thinking.
The reward for the computed next step is computed as the normalized edit distance between the generated action and the expert predicted step.
This assistance or supervision using the partially completed solutions and to only focus on generating the next step helps the LLM learn better.

The method requires a dataset or expert generations with an action-based formatting, where each example can be broken into a sequence of actions or steps.
The definition of action can vary with domain and datasets.
For this paper, the authors use step-wise structure of math solutions and environment commands for SWE tasks to distinguish individual actions.
The authors also use dynamic sampling to remove training examples with low variance in scores in all cases including baselines.

The authors show that the method performs well and provides a boost of 3% over RLVR on math tasks (3.7% for SRL+RLVR) on average. While naive RLVR does not yield any improvement, SRL does. SRL also results in 6.4% improvement in the oracle edit task and 4.4% in end-to-end task.

For qualitative analysis, the authors state that the method allows the agent multi-step thinking which manifests as, outlining all steps in a comprehensive initial plan, on-the fly adjustments, and reflective verification. However, only one representative example has been shown for this behavior. The overall answer lengths are also similar to the base-model.

**Strengths:**

- The method is simple and easy to follow.
- Based on the motivation, the approach of providing the supervision of first few steps and a dense reward makes sense.
- The method provides a boost in performance for math and SWE tasks.

**Weaknesses:**

- Adding comparisons against other methods that use intermediate rewards can be helpful (maybe process reward model (PRM) methods, e.g. [1]).
- Since the thinking patterns are listed as a contribution, the representative examples showing the interleaved thinking patterns and others can be expanded and analyzed.
- Since the method heavily relies on the data being in a specific format, there should be some discussion of limitations and expansion to other domains.


[1] Lu, J., Dou, Z., Wang, H., Cao, Z., Dai, J., Feng, Y. and Guo, Z., 2024. Autopsv: Automated process-supervised verifier. Advances in Neural Information Processing Systems, 37, pp.79935-79962.

**Questions:**

- Can the authors provide a comparison to some other self or expert supervised method that rewards intermediate steps?
- Did the authors try other reward schemes than string edit distance (like LLMs)? The same information can be presented in different sentences, and sometimes a slight difference in the text can drastically change the meaning; therefore, the edit-distance heuristic may not always hold.
- The temperature used (1.0) does not seem to be the default to my limited knowledge. If so, how do the average@32 scores change with temperature like 0.6/0.7?
- What was the final prompt used for inference, because the one mentioned in Appendix B states only one thinking step and outputs only one reasoning step (seems to be for training)?

---

> ### Author Response · Authors · 2025-12-02
> **Response to Reviewer iZLY**
>
> We thank the reviewer for their constructive feedback and for highlighting our motivation and the way to execute. We appreciate the opportunity to clarify our design and **have updated the manuscript with additional baselines, examples, and discussions in our updated PDF**.
>
> Below, we address each of your comments in detail.
>
> > **W1: Comparison against PRM methods:**
>
> We agree that Process Reward Models (PRMs) like AutoPSV are a relevant benchmark. However, we would like to highlight a key efficiency distinction:
>
> 1.**Computational Efficiency**: PRM-based methods require training and hosting a separate, often large, reward model to evaluate every step during generation/training. This introduces **significant latency and memory overhead**.
>
> 2. **SRL’s Approach**: A primary motivation for SRL is to achieve granular, step-wise supervision without the resource cost of an external PRM. By deriving intrinsic rewards from trajectory similarity, we **maintain a lightweight training loop**.
>
> We have tried to extend our framework to include PRM-based methods at the moment but have compute capacity difficulties. We will try to get more GPU resources to add experiments in the final version.
>
>
>
> > **W2: Expanded Analysis of Thinking Patterns**
>
> Thank you for this suggestion. We agree that the "interleaved thinking" capability is one of the core contributions that deserves more spotlight. We would like to introduce two detailed examples in the Appendix below:
> 1. **Appendix Example 2** highlights the model's ability to plan ahead, setting up a strategy before executing complex calculations.
> 2. **Appendix Example 3** demonstrates the model's ability to "think in between" steps. Unlike standard Chain-of-Thought (which usually dumps reasoning upfront), SRL allows the model to pause, reason, and self-correct in the middle of a trajectory, mimicking human-like problem-solving flow.
>
> > **W3: Data Formatting Dependencies and Generalization**
>
> We acknowledge the concern regarding data format dependencies. We would like to clarify that our framework **does not** require expert data to be **naturally pre-formatted** or manually decomposed. This is through our data-rewritten pipeline, which we detailed in the updated Appendix B.3. We employ an automated data re-writing pipeline to bridge the gap between raw expert data and SRL’s needs. The basic idea is
>
>
> (1.) We predefine a target step-wise format that is easy to parse
>
>
> (2.) We query a rewriter model (a teacher LLM) and we guide the rewriter to transform standard trajectories (correct answer, unstructured reasoning) into our target step-wise format.
>
> This pipeline allows SRL to ingest a wide range of existing datasets without them “naturally" fitting our step-wise setting.

---

> > ### Author Response · Authors · 2025-12-02
> > **Response to Reviewer iZLY (Cont.)**
> >
> > > **Q1: Comparison to other methods (included in the updated PDF)**
> >
> > We have implemented R$^3$ (arXiv:2402.05808) as an additional baseline to better contextualize SRL. Both methods use expert trajectories and reverse curriculum, but they differ fundamentally in their learning signal:
> > - R$^3$ (Outcome-Driven Exploration): Uses reverse curriculum to solve the sparse reward problem, helping the model find the correct answer. The reward is primarily based on the final outcome.
> >
> > - SRL (Process-Driven Supervision): Uses the trajectory to provide dense, intrinsic supervision. We teach the model how to reason at every step, not just where to end up.
> > Empirical Results: As shown in the table below, while R$^3$ performs well on easier cases, SRL provides a critical advantage on harder reasoning tasks (AIME 24/25) where step-by-step precision is paramount.
> >
> > | Model | AMC23 Avg@32 | AMC23 Greedy | AIME24 Avg@32 | AIME24 Greedy | AIME25 Avg@32 | AIME25 Greedy | Minerva Math | Average |
> > | :--- | :---: | :---: | :---: | :---: | :---: | :---: | :---: | :---: |
> > | *Base Model* | | | | | | | | |
> > | Qwen2.5-7B-Instruct | 49.3 | 50.0 | 10.5 | 13.3 | 7.5 | 6.7 | 34.9 | 24.6 |
> > | *Training with Outcome-Reward RL* | | | | | | | | |
> > | RL(VR) | 52.0 | 47.5 | 11.1 | 10.0 | 7.4 | 10.0 | 33.8 | 24.5 |
> > | SFT (outline) $\rightarrow$ RL(VR) | 37.6 | 35.0 | 4.9 | 3.3 | 4.5 | 6.7 | 30.1 | 17.4 |
> > | R$^3$ | **52.5** | **57.5** | 10.0 | 13.3 | 7.4 | 10.0 | 35.3 | 26.6 |
> > | *Training with SRL* | | | | | | | | |
> > | SRL | 51.5 | 50.0 | 13.2 | 16.7 | 7.1 | **13.3** | **36.4** | 27.6 |
> > | SRL $\rightarrow$ RLVR | 52.1 | **57.5** | **13.3** | **20.0** | **8.6** | 10.0 | **36.4** | **28.3** |
> >
> > Notably, SRL -> RLVR achieves 20.0% on AIME 24 (vs R$^3$'s 13.3%), suggesting that process-based supervision is a superior initialization for downstream RL compared to purely outcome-driven exploration.
> >
> >
> >
> > > **Q2: Alternative Reward Schemes (LLMs vs. Edit Distance)**
> >
> > We agree that string edit distance is a heuristic and that semantic equivalence is ideal. But we also want to highlight the points below:
> >
> > - **Speed**: We chose edit distance primarily for **training speed**. Querying an LLM reward model for every intermediate step during RL training is computationally prohibitive for our setting.
> > - **Future Work**: We agree that "LLM-as-a-Judge" is a promising direction for creating more robust semantic rewards, especially using strong teacher models. We have added a discussion on this in the new Section 5.4.
> >
> >
> > > **Q3: Temperature Settings of 1.0**
> >
> > We utilized a temperature of 1.0 for the Average@32 metric to align with the setting used in the DAPO paper (Yu et al., 2025). We have clarified this citation in the updated PDF.
> >
> >
> > > **Q4:  Inference Prompt**
> >
> > We apologize for the confusion. The prompt used for inference is indeed different from the training prompt to encourage full reasoning generation. We have explicitly included the inference prompt **in Appendix B.2.2** of the updated PDF.

---

### Official Review · Reviewer_JSqS · 2025-11-01

**Soundness:** 3
**Presentation:** 2
**Contribution:** 2
**Rating:** 2
**Confidence:** 3

**Summary:**

The SRL framework breaks down problems into a sequence of "actions". In each step, the model first generates an "internal monologue" and then executes an "action".

SRL's reward mechanism abandons sparse final-answer rewards and rigid imitation, instead providing dense signals at every step based on the similarity between the model's "action" and the expert's "action".

Crucially, SRL only rewards the "action," maintaining flexibility for the "internal monologue". This design combines dense signals (addressing the RLVR problem) with flexible reasoning (avoiding SFT rigidity) to promote stronger reasoning abilities.

Experiments show that SRL significantly outperforms SFT and RLVR on both mathematical reasoning and software engineering benchmarks. A curriculum learning strategy of "cold-starting with SRL then fine-tuning with RLVR" achieves the best results.

**Strengths:**

1. It finds a clever balance between SFT and RL. The core concept of "rewarding actions, liberating reasoning" is an important innovation for LLM reasoning training.
2. Comprehensive experiments were conducted on difficult benchmarks in mathematics and software engineering. The superior performance of SRL → RLVR is highly persuasive.
3. In-depth analysis (Section 5.2): High-quality ablation studies demonstrate that fine-grained guidance is key (Table 3). Behavioral analysis shows that the model exhibits more flexible, advanced reasoning patterns, rather than simply increasing output length (Figure 4).
4. Strong generalization ability: Successfully transferred from mathematics to software engineering tasks (Table 4), demonstrating its potential as a general-purpose reasoning framework.

**Weaknesses:**

1. Dependence on Expert Data and "Action" Definition: SRL is highly dependent on high-quality, easily decomposable expert trajectories and the definition of "actions."
2. Rationality of the Reward Function: The reward function is based on syntax (e.g., difflib) rather than common measures like KL divergence. How can the rationality of this approach be proven?
3. The primary weakness of this paper's methodology is its lack of novelty, as its core idea significantly overlaps with existing work: (1) Unoriginal Theoretical Foundation: Theoretically, the approach of using expert demonstrations to solve the sparse rewards problem is a classic paradigm in traditional reinforcement learning. Techniques such as Reverse Curriculum Learning and Learning from Demonstration have long been explored and validated in related fields. (2) Significant Overlap with Recent LLM Work: In the specific application of LLM reasoning, the proposed SRL framework is highly similar to the recent $R^3$ paper [1]. $R^3$ explicitly explored using expert demonstrations to train the model starting from intermediate states of the demonstration. Through a reverse curriculum, $R^3$ effectively converts sparse outcome rewards into "approximately step-by-step supervisory signals". The SRL paper similarly relies on decomposing expert trajectories into step-wise "actions" (what $R^3$ calls "intermediate states") and constructs partial trajectories for training. The core mechanisms of both methods are fundamentally alike. So, more discussion and comparison should be included.


[1] Training Large Language Models for Reasoning through Reverse Curriculum Reinforcement Learning

**Questions:**

1. Can you clarify the difference with works like $R^3$ in the paper?
2. Can you include more backbone models to make the paper more solid?

---

> ### Author Response · Authors · 2025-12-02
> **Response to Reviewer JSqS**
>
> We sincerely thank you for your thoughtful review. We are encouraged that you found our framework finds a clever balance between SFT and RL! Below, we provide detailed responses to each of your comments and hope to address any further considerations you may have.
>
>
> > **W1: Dependence on Expert Data and "Action" Definition**
>
> We thank the reviewer for this insightful comment. We agree that the availability of high-quality, decomposable expert trajectories is critical for SRL. However, we would like to clarify that our framework does not strictly require expert data to be *naturally pre-formatted* in a step-wise manner, nor does it require manual decomposition.
> To address this dependency, we employ an automated **data re-writing pipeline** that significantly relaxes the formatting constraints on the initial expert data. **As detailed in the updated Appendix B.3**, our approach is as follows:
> - **Automated Formatting**: We define a target step-wise format that is easy to parse. We then leverage a stronger model (e.g., a large-scale LLM) to act as a "rewriter."
> - **Two-Shot Prompting**: Using just two-shot examples to clarify the instructions, we query the rewriter to transform standard expert trajectories (which contain correct answers but unstructured reasoning) into our desired step-by-step format.
> This process acts as a bridge, allowing SRL to utilize a wide range of existing expert data without requiring them to "naturally" fit our action definition in the first trial.
> In short, while the *definition* of an action remains important, the burden of obtaining data that adheres to it is handled by the model-driven rewriting process rather than manual curation. We have updated the paper to include these full implementation details.
>
> > **W2: Rationality of the Reward Function against KL Divergence**
>
>
> We thank the reviewer for this insightful question regarding the theoretical grounding of our reward function. We appreciate the opportunity to clarify why we chose a syntax-based similarity measure (like difflib) over probabilistic measures like KL divergence for our implementation.
> The rationality are highlighted below:
> 1. **KL Divergence on token enforces rigid mimicry (SFT)**: Standard SFT effectively minimizes the forward KL divergence between the expert distribution and the model distribution. While theoretically sound, in practice, this forces the model to perform "rigid token mimicry"—penalizing any deviation from the expert's exact phrasing. Our goal with SRL was specifically to **break this rigidity**.
> 2. **Syntax-based rewards provide "outcome-oriented" dense supervision**: Unlike KL divergence, which operates on probability distributions, our syntax-based reward operates on the **generated text itself**. This also provides the advantage and easiness of assigning **Partial Credit**. It provides a dense, scalar signal that reflects how close the generated action is to the expert's, *even if the exact token probabilities differ*. This is akin to a "soft" correctness check that is more granular than a binary pass/fail (RLVR) but less constraining than per-token likelihood (SFT).
> 3. **Empirical Verification**: We believe the "rationality" of this reward design is best evidenced by its empirical performance. As shown in our experiments, SRL significantly outperforms SFT (which optimizes the standard probabilistic objective) and RLVR (which uses sparse rewards). This suggests that for complex step-wise reasoning, a flexible, syntax-guided reward effectively bridges the gap between imitation and exploration.
> We hope this clarifies our choice of reward function.

---

> ### Author Response · Authors · 2025-12-02
> **Response to Reviewer JSqS (Cont.)**
>
> > **W3 & Q1: Novelty and Comparison with $R^3$**
>
> We appreciate the opportunity to clarify the positioning of our work relative to classical Learning from Demonstration and recent curriculum  approaches like $R^3$.
> While we agree that both methods utilize expert trajectories and a reverse curriculum structure, we respectfully argue that the **nature of the supervision** and the **learning objective** differ fundamentally. This distinction is not merely theoretical; as our new results on challenging benchmarks (AIME 24/25) demonstrate, it leads to significant differences in performance on complex reasoning tasks as we shown in the below additional experimental results:
>
> | Model | AMC23 Avg@32 | AMC23 Greedy | AIME24 Avg@32 | AIME24 Greedy | AIME25 Avg@32 | AIME25 Greedy | Minerva Math | Average |
> | :--- | :---: | :---: | :---: | :---: | :---: | :---: | :---: | :---: |
> | *Base Model* | | | | | | | | |
> | Qwen2.5-7B-Instruct | 49.3 | 50.0 | 10.5 | 13.3 | 7.5 | 6.7 | 34.9 | 24.6 |
> | *Training with Outcome-Reward RL* | | | | | | | | |
> | RL(VR) | 52.0 | 47.5 | 11.1 | 10.0 | 7.4 | 10.0 | 33.8 | 24.5 |
> | SFT (outline) $\rightarrow$ RL(VR) | 37.6 | 35.0 | 4.9 | 3.3 | 4.5 | 6.7 | 30.1 | 17.4 |
> | R$^3$ | **52.5** | **57.5** | 10.0 | 13.3 | 7.4 | 10.0 | 35.3 | 26.6 |
> | *Training with SRL* | | | | | | | | |
> | SRL | 51.5 | 50.0 | 13.2 | 16.7 | 7.1 | **13.3** | **36.4** | 27.6 |
> | SRL $\rightarrow$ RLVR | 52.1 | **57.5** | **13.3** | **20.0** | **8.6** | 10.0 | **36.4** | **28.3** |
>
> We would like to highlight some comparison here:
>
> **1. Distinction in Theoretical Mechanism: Outcome v.s Process**
>
> **$R^3$ (Outcome-Driven)**: $R^3$ starts training near the end of a trajectory to help the model get the correct answer more frequently. The reward signal is **still binary** and delayed—the model is rewarded if it reaches the correct final answer. The curriculum acts as a heuristic to facilitate exploration, but the model is still optimizing for the outcome, not necessarily the quality of the intermediate reasoning steps.
>
> **SRL (Process-Driven)**: In contrast, SRL uses the trajectory to provide **dense, intrinsic supervision** via action similarity rewards. We are not merely guiding the model to the correct answer; we are explicitly supervising **how it reasons at every step**. By rewarding the similarity between the model's generated action and the expert's action, SRL provides a gradient signal even when the final answer is incorrect or not yet reached. This effectively bridges the gap between the rigid mimicry of SFT (Behavior Cloning) and the sparse signals of RL.
>
>
> **2. Empirical Comparison on Hard Reasoning Tasks**
>
> As shown in the table above, SRL demonstrates a critical advantage on harder tasks (like AIME24, AIME25) while some easier tasks like AMC23 and Minera matches the results of $R^3$
>
>
> > **Q2: Scalability and Generalizability to Different Model**
>
> **Experimental Results on Qwen-2.5-3B**
>
> To test the robustness of our framework across different model scales, we extended our experiments to the **Qwen-2.5-3B-Instruct** model. The results on the 3B model strongly support the consistency of our method.
> As shown in the table below, SRL combined with RLVR yields a consistent performance boost, improving the average accuracy by 3.1 absolute percentage points over the base model and surpassing standard RL(VR).
>
>
> | Model | AMC23 Avg@32 | AMC23 Greedy | AIME24 Avg@32 | AIME24 Greedy | AIME25 Avg@32 | AIME25 Greedy | Minerva Math | Average |
> | :--- | :---: | :---: | :---: | :---: | :---: | :---: | :---: | :---: |
> | Qwen2.5-3B-Instruct | 32.5 | 45.0 | 3.3 | 3.3 | 1.7 | 3.3 | 25.7 | 16.4 |
> | RL(VR) | 35.0 | 42.5 | 6.7 | 3.3 | 6.7 | 3.3 | 26.5 | 17.7 |
> | SRL → RLVR | 37.5 | 47.5 | 6.7 | 3.3 | 6.7 | 6.7 | 28.3 | 19.5 |
>
> ---
> **We have updated Section 5.1** to explicitly discuss the relationship with $R^3$, clarifying these methodological differences, and includes the results of different model.

---

### Official Review · Reviewer_CWA3 · 2025-11-01

**Soundness:** 3
**Presentation:** 3
**Contribution:** 3
**Rating:** 6
**Confidence:** 4

**Summary:**

This paper proposes Supervised Reinforcement Learning (SRL), a new framework that bridges the gap between Supervised Fine-Tuning (SFT) and Reinforcement Learning with Verifiable Rewards (RLVR) for complex reasoning tasks.
This approach provides richer learning signals than RLVR and avoids the rigid imitation of SFT.
Experiments on mathematical reasoning and software engineering benchmarks show that SRL significantly outperforms both baselines, and a curriculum of SRL followed by RLVR achieves the best overall results.

**Strengths:**

1. The novel SRL framework addresses RLVR's sparse rewards and SFT's rigid mimicry by decomposing expert trajectories into step-wise logical actions with dense sequence-similarity rewards, uniquely combining SFT’s supervision and RL’s strengths and showing superior performance on math reasoning benchmarks.
2. SRL demonstrates robust cross-domain efficacy across different multi-step reasoning tasks, proving its versatility.
3. SRL excels in data efficiency for hard problems, using limited expert data to avoid SFT's performance degradation and enabling a high-performing SRL→RLVR curriculum.

**Weaknesses:**

1. SRL relies on the decomposition of expert trajectories into logical actions without providing details on automation for unstructured trajectories. Therefore, I'm not sure if it is scalable for large-scale or unstructured tasks and whether it introduces subjective bias.
2. The reward only considers action similarity, potentially leading to false positive rewards from flawed reasoning. This impact should be considered.
3. The experiments are conducted solely on small to mid-sized models. It would strengthen the paper to include results or discussion on whether SRL’s advantages extend to larger model sizes, as this would provide a clearer picture of its scalability and general applicability.

**Questions:**

1. I am curious how robust your difflib-based action-similarity reward is to “near-miss” actions
2. I think the framework genuinely novel and potentially a new training paradigm for leveraging reward signals. I encourage you to evaluate SRL beyond structured domains in your paper.

---

> ### Author Response · Authors · 2025-12-02
> **Response to Reviewer CWA3**
>
> We sincerely thank you for your thoughtful review. We are encouraged that you found our framework novel and generalized! Below, we provide detailed responses to each of your comments and hope to address any further considerations you may have.
>
> > **W1: How to decompose expert reasoning traces?**
>
> We conduct the task decomposition via a data re-writing process. We first decide a format to rewrite, where the format is easy to parse trajectory into steps. Then based on the designed format, we query large models to rewrite its full trajectory of answer (known to be correct answer) into a step-by-step format with two-shot examples to ensure the instructions are clear. The full details have been updated in the new version of PDF in Appendix B.3. Through this data re-write process, we turn a task its a multi-step reasoning process into data SRL can use
>
> > **W2: False positive rewards from flawed reasoning**
>
> We appreciate this insightful observation. We agree that SRL, like other imitation-based methods, can bias the model toward the potential flawed reasoning path from the teacher.
>
> However, we address this limitation through our proposed curriculum: SRL followed by RLVR. SRL provides the necessary grounding and structure for complex reasoning (which pure RL struggles to discover from scratch), while the subsequent RLVR stage allows the model to optimize its policy based on verification feedback. As shown in Table 1, the performance gains from adding RLVR suggest that the model successfully generalizes beyond the teacher's exact trajectory, effectively "unlearning" unnecessary detours once the task is grounded. While heuristic data filtering could further improve SRL efficiency, our results demonstrate that the SRL+RLVR paradigm is robust.
>
> > **W3: Scalability and Generalizability to Different Model Sizes**
>
> We thank the reviewer for this insightful suggestion. We agree that understanding the scalability of SRL is crucial for establishing its general applicability.
>
> **1.Experimental Results on Qwen-2.5-3B**
>
> To test the robustness of our framework across different model scales, we extended our experiments to the **Qwen-2.5-3B-Instruct** model. While we initially attempted to train **Qwen-2.5-14B-Instruct**, we encountered computational resource constraints (GPU memory limitations) that prevented full convergence within the rebuttal period. However, the results on the 3B model strongly support the consistency of our method.
> As shown in the table below, SRL combined with RLVR yields a consistent performance boost, improving the average accuracy by 3.1 absolute percentage points over the base model and surpassing standard RL(VR).
>
>
> | Model | AMC23 Avg@32 | AMC23 Greedy | AIME24 Avg@32 | AIME24 Greedy | AIME25 Avg@32 | AIME25 Greedy | Minerva Math | Average |
> | :--- | :---: | :---: | :---: | :---: | :---: | :---: | :---: | :---: |
> | Qwen2.5-3B-Instruct | 32.5 | 45.0 | 3.3 | 3.3 | 1.7 | 3.3 | 25.7 | 16.4 |
> | RL(VR) | 35.0 | 42.5 | 6.7 | 3.3 | 6.7 | 3.3 | 26.5 | 17.7 |
> | SRL → RLVR | 37.5 | 47.5 | 6.7 | 3.3 | 6.7 | 6.7 | 28.3 | 19.5 |
>
> **2. Discussion on Scalability**
> These results, combined with our main paper’s findings on 7B models, suggest that SRL is effective across varying model capacities. Regarding larger models (e.g., 14B+), we posit that SRL’s effectiveness is less about model size per se and more about the **difficulty of the data relative to the model's capability**.
> SRL is designed to bridge the gap in "hard" reasoning tasks where the model struggles to explore successful trajectories (the "cold start" problem in RL). As model size increases, the frontier of what constitutes a "hard" problem shifts. We hypothesize that SRL will remain highly effective for larger models provided the target tasks are sufficiently challenging that the base model cannot easily solve them via standard exploration or SFT alone.
>
> ---
> We have added these new results and the discussion on scalability to Section 5.1 of the revised PDF.

---

> > ### Author Response · Authors · 2025-12-02
> > **Response to Reviewer CWA3 (Cont.)**
> >
> > > **Q1: Robustness of difflib-based action-similarity reward to "near-miss" actions**
> >
> > Thank you for raising this important point. We specifically chose the difflib-based reward (using Python's standard difflib.SequenceMatcher) precisely because it offers a granular, continuous signal that is robust to minor variations, unlike a binary exact-match reward. This function is specifically useful for tasks with “structures” such as code editing of equations.
> >
> >
> > To elaborate, difflib implements the Ratcliff-Obershelp algorithm, which recursively finds the longest contiguous matching subsequences between the model's action and the expert's action. The similarity score ($S$) is calculated as:
> > $$S = \frac{2 \cdot M}{T}$$
> > Where $M$ is the number of matching characters and $T$ is the total number of characters in both sequences.
> > Why this ensures robustness on near-miss:
> > - **Smooth Gradient**: A "near-miss" action—such as one with a typo, a slightly different variable name, or minor syntactic variance—will still share a high number of matching characters ($M$) with the expert action. Consequently, the reward will be high (e.g., 0.85 or 0.90) rather than 0.
> > - **Partial Credit**: This mechanism naturally assigns "partial credit," guiding the model toward the correct format and content without penalizing it strictly for trivial deviations. This dense, smooth reward landscape helps the model converge faster than sparse binary rewards would allow.
> > Yet, we also think that the way to calculate similarity reward functions can be further explored in different tasks.
> >
> > > **Q2: Evaluating SRL beyond structured domains**
> >
> > While our current work focuses on Math and Software Engineering where ground truth actions are often clearer, we believe SRL has significant potential in broader, less structured contexts because of its flexibility.
> >
> > As we expand into open-ended domains (e.g., creative writing or report writing), exact character-level similarity may become too rigid. We envision two primary "reward shaping" extensions to adapt SRL for these tasks:
> >
> > 1. Embedding-based Rewards: Using lightweight semantic embeddings to calculate cosine similarity between the model's step and the expert's step, allowing for rewording that preserves meaning.
> >
> > 2. Teacher LLM as Judge: employing a Teacher LLM to score the semantic equivalence of a step, providing a more nuanced reward signal that captures intent rather than just surface form.
> >
> > We are excited to explore these directions in future work to generalize the SRL paradigm further.

---

### Official Review · Reviewer_VwHa · 2025-11-02

**Soundness:** 3
**Presentation:** 3
**Contribution:** 3
**Rating:** 4
**Confidence:** 4

**Summary:**

This paper introduces Supervised Reinforcement Learning, a framework designed to enhance complex reasoning in LLMs by decomposing problems into step-wise actions. SRL combines dense, step-level rewards with flexibility in internal reasoning generation. The method is evaluated on mathematical reasoning and software engineering tasks, showing improvements over SFT and RL baselines. Key contributions include: (1) a step-wise reward signal to mitigate sparse rewards, (2) empirical gains on challenging benchmarks, and (3) a curriculum strategy (SRL→RLVR) for further refinement.

**Strengths:**

The problem formulation (e.g., action-based decomposition) is well-motivated and intuitive.
Results on math benchmarks show consistent improvements, with ablations justifying key components.
The curriculum strategy (SRL→RLVR) and flexibility in internal monologue generation are valuable insights.

**Weaknesses:**

The methodology bears significant resemblance to R³ (arXiv:2402.05808), which also uses a reverse curriculum with intermediate state sampling and outcome supervision for step-wise learning. Though SRL uses action similarity rewards instead of final-answer rewards, the high-level strategy of "starting from expert states and moving backward" is not sufficiently differentiated.
Comparisons focus on SFT and RLVR but omit advanced RL methods (e.g., DAPO, Dr. GRPO) and direct comparisons to R³.

**Questions:**

None

---

> ### Author Response · Authors · 2025-12-02
> **Response to Reviewer VwHa**
>
> We sincerely thank you for your thoughtful review. We are encouraged that you found our problem formulation intuitive and our curriculum strategy valuable! Your feedback regarding the relationship with R$^3$ and comparisons to advanced RL methods is excellent, and it has motivated us to run additional experiments to strengthen our claims.
> We have addressed your concerns below:
>
>
> **1. Differentiating SRL from R$^3$: Process vs. Outcome**
>
> We appreciate you pointing out the structural similarities with R$^3$ (arXiv:2402.05808). You are correct that both methods leverage expert trajectories and a reverse curriculum. However, we believe there is a fundamental distinction in how that data is utilized, which leads to significant performance differences on complex tasks.
>
> **R$^3$ focuses on Exploration (Outcome-Driven)**: R$^3$ uses the reverse curriculum primarily to solve the sparse reward problem. By starting near the end, it helps the model "find" the correct answer more often. However, the reward signal remains outcome-based. The model is rewarded for reaching the answer, not necessarily for the reasoning steps taken to get there.
>
> **SRL focuses on Supervision (Process-Driven)**: In contrast, SRL uses the trajectory to provide dense, intrinsic supervision via action similarity rewards. We are not just teaching the model "where to end up," but "how to reason" at every single step.
> New Empirical Evidence: To make this distinction concrete, we implemented R$^3$ as a baseline. As shown in the table below, while R$^3$ is highly effective on standard benchmarks (AMC23), SRL provides a critical advantage on harder benchmarks (AIME 24/25):
>
>
> | Model | AMC23 Avg@32 | AMC23 Greedy | AIME24 Avg@32 | AIME24 Greedy | AIME25 Avg@32 | AIME25 Greedy | Minerva Math | Average |
> | :--- | :---: | :---: | :---: | :---: | :---: | :---: | :---: | :---: |
> | *Base Model* | | | | | | | | |
> | Qwen2.5-7B-Instruct | 49.3 | 50.0 | 10.5 | 13.3 | 7.5 | 6.7 | 34.9 | 24.6 |
> | *Training with Outcome-Reward RL* | | | | | | | | |
> | RL(VR) | 52.0 | 47.5 | 11.1 | 10.0 | 7.4 | 10.0 | 33.8 | 24.5 |
> | SFT (outline) $\rightarrow$ RL(VR) | 37.6 | 35.0 | 4.9 | 3.3 | 4.5 | 6.7 | 30.1 | 17.4 |
> | R$^3$ | **52.5** | **57.5** | 10.0 | 13.3 | 7.4 | 10.0 | 35.3 | 26.6 |
> | *Training with SRL* | | | | | | | | |
> | SRL | 51.5 | 50.0 | 13.2 | 16.7 | 7.1 | **13.3** | **36.4** | 27.6 |
> | SRL $\rightarrow$ RLVR | 52.1 | **57.5** | **13.3** | **20.0** | **8.6** | 10.0 | **36.4** | **28.3** |
>
>
> Notably, SRL alone achieves 16.7% on AIME 24 (vs R$^3$'s 13.3%). When used as a warm-up for online RL, SRL $\rightarrow$ RLVR hits 20.0%, suggesting that process-based supervision is a superior initialization for downstream RL compared to purely outcome-driven exploration.
>
>
> **2. Comparisons to Advanced RL Methods:**
>
> Thank you for suggesting comparisons to methods like DAPO and Dr. GRPO. We fully agree that contextualizing SRL against the state-of-the-art is important.
>
> Yet we also want to clarify that SRL is a framework, not just an optimizer: We view SRL as a data-centric learning framework (creating reasoning priors from expert data) rather than a direct competitor to RL optimization algorithms like DAPO or Dr. GRPO..
>
> SRL is orthogonal to the choice of the RL algorithm used. For example: One could essentially run SRL $\rightarrow$ GRPO or SRL $\rightarrow$ DAPO. Our results demonstrate that SRL provides a strictly better starting point than SFT, regardless of the downstream optimizer used.
>
>
> Additionally, we are glad for the opportunity to clarify our "RLVR" baseline. As detailed in Section 5.1, our implementation isn't naive; it incorporates dynamic sampling strategies akin to DAPO. This ensures that when we show SRL improves over RLVR, we are comparing against a robust, modern outcome-reward baseline.
>
> ----
> We hope these new results and clarifications address your concerns! We have updated the final PDF to include the R$^3$ direct comparison and a more detailed discussion on our baselines (Section 5.1)

---

### Official Review · Reviewer_WNpC · 2025-11-02

**Soundness:** 4
**Presentation:** 3
**Contribution:** 3
**Rating:** 6
**Confidence:** 4

**Summary:**

This paper introduces Supervised Reinforcement Learning, a framework that bridges the gap between Reinforcement Learning with Verifiable Rewards, which suffers from sparse rewards on difficult problems, and Supervised Fine-Tuning, which often leads to overfitting through rigid token mimicry. SRL reformulates problem-solving as a sequence of logical actions, where the model generates an internal reasoning monologue followed by an action, receiving dense rewards based on action similarity to expert demonstrations. This allows flexibility in reasoning while providing granular feedback.

**Strengths:**

1. The paper presents a well-motivated and innovative approach to addressing limitations in LLM training for complex reasoning, with clear writing.

2. The introduction of SRL creatively combines elements of imitation learning and RL by decomposing expert trajectories into step-wise actions and rewarding only the actions, allowing the model to develop its own internal monologues. This novel formulation addresses the sparse reward issue in RLVR and the rigid mimicry in SFT, drawing from prior work like GRPO but extending it to dense, sequence-similarity-based rewards for hard problems.

3. By enabling effective learning on difficult datasets like s1K, SRL has broad implications for enhancing LLMs in domains requiring multi-step reasoning, such as math and software engineering. Its generalization to agentic tasks and potential as a cold-start for RLVR pipelines could influence future distillation and RL methods for smaller models.

**Weaknesses:**

1. The method lacks sufficient implementation details, raising concerns about reproducibility. For instance, the prompts or heuristics used to decompose expert reasoning traces into individual steps (e.g., parsing numbered sections from DeepSeek R1 outputs) are not provided, making it challenging to replicate the step-wise data construction process.

2. Although SRL rewards only the action sequence to avoid token-by-token memorization, this still imposes constraints on the model's output. For example, it may force the student to mimic potentially suboptimal or erroneous intermediate actions from the teacher (e.g., steps involving reflection to correct mistakes), even if the student could compute them correctly without such detours. This could limit flexibility in cases where the teacher's reasoning includes unnecessary self-corrections.

3. Experiments are limited to a single 7B model (Qwen2.5-7B-Instruct), without exploring scalability across different model sizes (e.g., 1B, 13B, or larger). This leaves uncertainty about whether SRL's benefits hold for smaller or larger models, potentially restricting insights into its robustness.

**Questions:**

1. For extremely difficult problems where the model fails to rollout any reasonable intermediate step actions (e.g., pass@k near zero even for steps), how does SRL enable effective learning? Could the dense rewards still provide guidance, or would additional techniques like bootstrapping be needed?

2. The method relies on a strong teacher model (e.g., DeepSeek R1) to generate reliable expert trajectories. If no such trustworthy teacher is available (e.g., for novel domains), how could SRL be adapted—perhaps through self-generated or synthetic data, or iterative refinement?

---

> ### Author Response · Authors · 2025-12-02
> **Response to Reviewer WNpC (Part 1)**
>
> Thanks for your positive review and constructive comments. We appreciate your recognition of SRL’s clear motivation and our key design of extending rewards to dense, sequence-similarity-based rewards for hard problems. Below, we provide detailed responses to each of your comments and hope to address any further considerations you may have.
>
> > **W1: How to decompose expert reasoning traces?**
>
> We conduct the task decomposition via a data re-writing process. We first decide a format to rewrite, where the format is easy to parse trajectory into steps. Then based on the designed format, we query large models to rewrite its full trajectory of answer (known to be correct answer) into a step-by-step format with two-shot examples to ensure the instructions are clear. The full details have been updated in the new version of PDF in Appendix B.3.
>
> > **W2: SRL may force the student to mimic suboptimal or erroneous intermediate actions.**
>
> We appreciate this insightful observation. We agree that SRL, like other imitation-based methods, can bias the model toward the teacher's specific reasoning path, including potential inefficiencies like self-correction.
>
> However, we mitigate this limitation through our proposed curriculum: SRL followed by RLVR. SRL provides the necessary grounding and structure for complex reasoning (which pure RL struggles to discover from scratch), while the subsequent RLVR stage allows the model to optimize its policy based on verification feedback. As shown in Table 1, the performance gains from adding RLVR suggest that the model could mitigate the potential suboptimal learning.. While heuristic data filtering could further improve SRL efficiency, our results demonstrate that the SRL+RLVR paradigm is robust.
>
> > **W3: Exploring scalability across different model sizes.**
>
> We conduct additional experiment on Qwen-2.5-3B-Instruct and provide the results below:
>
> | Model | AMC23 Avg@32 | AMC23 Greedy | AIME24 Avg@32 | AIME24 Greedy | AIME25 Avg@32 | AIME25 Greedy | Minerva Math | Average |
> | :--- | :---: | :---: | :---: | :---: | :---: | :---: | :---: | :---: |
> | Qwen2.5-3B-Instruct | 32.5 | 45.0 | 3.3 | 3.3 | 1.7 | 3.3 | 25.7 | 16.4 |
> | RL(VR) | 35.0 | 42.5 | 6.7 | 3.3 | 6.7 | 3.3 | 26.5 | 17.7 |
> | SRL → RLVR | 37.5 | 47.5 | 6.7 | 3.3 | 6.7 | 6.7 | 28.3 | 19.5 |
>
> We can see that our method yields a consistent performance boost, improving the average score over base model by 3.1% and surpassing RL(VR). **We have included the results and discussion into our new version of PDF in Section 5.1**.
>
> We have tried to extend our framework to Qwen-2.5-14B-Instruct at the moment but have compute capacity difficulties. We will try to get more GPU resources to add experiments on larger sizes of models like Qwen-2.5-14B-Instruct in the final version.

---

> ### Author Response · Authors · 2025-12-02
> **Response to Reviewer WNpC (Part 2)**
>
> > **Q1: Extremely difficult steps**
>
> We thank the reviewer for this critical question. We explicitly designed SRL to address the “cold start” problem where pass@k is near zero even in steps. We would like to highlight SRL’s strategy to potentially enable effective learning in these cases:
>
> - **Implicit Curriculum via Dynamic Sampling**:
> As noted in our draft, we employ dynamic sampling to filter out steps data where the reward standard deviation is low (implying the model is either completely lost or fully confident). This naturally focuses training on the "boundary" steps where the model is currently struggling but capable of improvement. This creates an auto-curriculum: the model first consolidates simpler steps (which then become low-variance and are filtered out) before graduating to learning the more difficult intermediate reasoning steps.
> - **Dense Step-wise Supervision (Breaking the Dependency on Final Success)**:
> Unlike outcome-based methods that require a full successful rollout to generate a learning signal, SRL provides dense rewards at each step by measuring similarity to the expert's action. Even if a rollout eventually fails (resulting in 0 final reward), the model still receives strong positive reinforcement for the correct intermediate steps it did take. This allows the model to "climb" the expert trajectory on easier steps at a time, creating better usage of the data.
>
>
> > **Q2: How could be the case where no such trustworthy teacher is available**
>
> We thank the reviewer for this forward-looking question. In novel domains where a strong teacher is absent, we propose adapting SRL via Iterative Self-Bootstrapping, effectively turning the model into its own teacher. We have added a brief discussion on this in Section 5.4 of the revision. Our thought with this question is outlined below:
>
> 1. Bootstrapping from Verifiable Outcomes (STaR-like approach): In domains like math or coding, we can often verify the final outcome (via compilers or rule-checkers) even if we lack the reasoning steps. SRL can be adapted by generating multiple rollouts from the current best model (e.g., via rejection sampling) and filtering for correctness. The successful trajectories—initially rare—can be treated as “pseudo-expert” data. SRL then extracts the step-wise reasoning from these successful instances to stabilize training, progressively improving the policy to generate such trajectories more frequently.
>
> 2. Trajectory Refinement via Summarization: Self-generated trajectories are often noisy or redundant compared to distilled expert traces. As noted in our data preparation response above, we can additionally employ a "Summarize-and-Prune" step to prepare the data to condense lengthy trajectory into clean, logical steps before using them for SRL supervision. This ensures the "step-wise" reward signal remains high-quality.

---

### Author Response · Authors · 2025-12-02
**Summary of changes reflecting all the reviews**

Dear Area Chair and Reviewers,

We sincerely thank the reviewers for their constructive feedback. In response, we have significantly revised our paper (changes highlighted in purple-red), added new baselines ($R^3$), extended model scalability experiments, and clarified data processing pipelines and other details.

Below is a summary of our rebuttal updates to assist the Area Chair in their assessment.

---

**Summary of Strengths:**

We appreciate that the reviewers unanimously recognized the core value of our work:


- **Methodological Innovation**: Reviewers `WNpC`, `CWA3`, and `JSqS` highlighted SRL as a novel framework that effectively balances the strengths of SFT (imitation) and RL (exploration), solving the "rigid mimicry" and "sparse reward" problems.

- **Empirical Rigor**: Reviewers `WNpC`, `VwHa`, `CWA3`, `JSqS`, and `iZLY` commended the method's superior performance on challenging math and SWE benchmarks, supported by in-depth behavioral analysis.

- **Generalization & Impact**: The cross-domain efficacy (Math & SWE) and the proposed curriculum strategy (SRL $\to$ RLVR) were noted as valuable insights for future reasoning pipelines (Reviewers `WNpC`, `VwHa`, `CWA3`, `JSqS`, `iZLY`).

---

**Summary of Clarifications and Additional Experiments:**


1. **New Baseline Comparison: SRL vs. $R^3$** (Response to `WNpC`, `VwHa`, `JSqS`, `iZLY` )

We implemented the requested $R^3$ baseline [1] in updated **Section 5.1**. While both methods utilize expert trajectories, they fundamentally differ in their learning signal:

- **R$^3$ focuses on Exploration (Outcome-Driven)**: Uses a reverse curriculum to help the model "find" the answer. It rewards reaching the destination.


- **SRL focuses on Supervision (Process-Driven)**: Uses action similarity rewards to provide dense supervision. It teaches the model how to reason at every step.


The empirical evidence is summarized in the below table: While R$^3$ is highly effective on standard benchmarks (AMC23), SRL offers a critical advantage on complex reasoning tasks (AIME 24/25), proving that process-based supervision is a superior initialization for downstream RL.


| Model | AMC23 Avg@32 | AMC23 Greedy | AIME24 Avg@32 | AIME24 Greedy | AIME25 Avg@32 | AIME25 Greedy | Minerva Math | Average |
| :--- | :---: | :---: | :---: | :---: | :---: | :---: | :---: | :---: |
| *Base Model* | | | | | | | | |
| Qwen2.5-7B-Instruct | 49.3 | 50.0 | 10.5 | 13.3 | 7.5 | 6.7 | 34.9 | 24.6 |
| *Training with Outcome-Reward RL* | | | | | | | | |
| RL(VR) | 52.0 | 47.5 | 11.1 | 10.0 | 7.4 | 10.0 | 33.8 | 24.5 |
| SFT (outline) $\rightarrow$ RL(VR) | 37.6 | 35.0 | 4.9 | 3.3 | 4.5 | 6.7 | 30.1 | 17.4 |
| R$^3$ | **52.5** | **57.5** | 10.0 | 13.3 | 7.4 | 10.0 | 35.3 | 26.6 |
| *Training with SRL* | | | | | | | | |
| SRL | 51.5 | 50.0 | 13.2 | 16.7 | 7.1 | **13.3** | **36.4** | 27.6 |
| SRL $\rightarrow$ RLVR | 52.1 | **57.5** | **13.3** | **20.0** | **8.6** | 10.0 | **36.4** | **28.3** |


**2. Model Scalability** (Response to `WNpC`, `CWA3`, `JSqS` )

We extended our evaluation to include the Qwen-2.5-3B-Instruct model. SRL consistently improves upon the base model, demonstrating that our method is effective across model scales (**updated Section 5.1**)

| Model | AMC23 Avg@32 | AMC23 Greedy | AIME24 Avg@32 | AIME24 Greedy | AIME25 Avg@32 | AIME25 Greedy | Minerva Math | Average |
| :--- | :---: | :---: | :---: | :---: | :---: | :---: | :---: | :---: |
| Qwen2.5-3B-Instruct | 32.5 | 45.0 | 3.3 | 3.3 | 1.7 | 3.3 | 25.7 | 16.4 |
| RL(VR) | 35.0 | 42.5 | 6.7 | 3.3 | 6.7 | 3.3 | 26.5 | 17.7 |
| SRL → RLVR | 37.5 | 47.5 | 6.7 | 3.3 | 6.7 | 6.7 | 28.3 | 19.5 |


**3. Clarification on Data Preparation** (Response to `WNpC`, `CWA3`, `JSqS`, `iZLY` )

We clarified **in Section 5 and Appendix B.3** that SRL does not require naturally step-wise data. We utilize a scalable LLM-based rewriting pipeline to decompose existing expert trajectories (using 2-shot prompting) into the format SRL requires. This allows SRL to ingest a wide range of existing datasets without manual annotation.


**4. Evaluation Protocols Details:** (Response to `iZLY`)

We have expanded on the evaluation details **in Section 5.1 and Appendix B.2** to ensure reproducibility and clarity.


**5. Limitations & Future Work Discussion on extending to other domain:** (Response to `WNpC`, `CWA3`, `iZLY`)

We further added Section 5.4 to discuss extensions to open-ended domains (e.g., creative writing), proposing embedding-based rewards or "LLM-as-a-Judge" to replace exact matching in future iterations.

---
We believe these additions robustly address the reviewers' concerns and strengthen the paper's contribution.

Best,

The Authors

---

Reference:

[1] Xi, Z., Chen, W., Hong, B., Jin, S., Zheng, R., He, W., … Huang, X. (2024). Training Large Language Models for Reasoning through Reverse Curriculum Reinforcement Learning. Forty-First International Conference on Machine Learning, ICML 2024, Vienna, Austria, July 21-27, 2024

---

### Meta-Review · Area_Chair_3WNw · 2025-12-21

**Summary:**

This paper introduces Supervised Reinforcement Learning, a framework designed to bridge the gap between supervised fine-tuning and reinforcement learning with verifiable rewards. By decomposing expert trajectories into logical actions, the model generates internal reasoning monologues followed by specific actions that receive dense rewards based on similarity to expert demonstrations. This approach promotes flexible reasoning while addressing sparse reward issues, demonstrating significant performance gains across mathematical reasoning and agentic software engineering benchmarks.

**Strengths:**
1. The framework provides a novel and well-motivated approach that effectively balances the strengths of imitation learning and reinforcement learning. By rewarding only actions while liberating internal monologues, it successfully addresses both the sparse reward problem and the issue of rigid token mimicry.
2. Extensive experiments on challenging mathematical reasoning and software engineering benchmarks demonstrate that the method consistently outperforms standard SFT and RLVR baselines. The proposed curriculum strategy starting with SRL and followed by RLVR achieves particularly strong results across diverse tasks.
3. The method exhibits robust cross-domain efficacy and generalizes well to agentic tasks, showing its potential as a versatile training paradigm. It also demonstrates high data efficiency, particularly on difficult problems where traditional methods often struggle or degrade in performance.

**Weaknesses:**
1. Several reviewers noted significant overlap with existing work like R3 and traditional reverse curriculum learning, suggesting that the core theoretical foundation lacks sufficient novelty.
2. The current methodology lacks specific details regarding the automation of step-wise data construction and the heuristics used for parsing expert trajectories, raising reproducibility and scalability concerns, and potential subjective bias.
3. Experiments are limited to small or mid-sized models, leaving uncertainty about whether the benefits of this framework scale effectively to larger architectures.

**Reviewer Concerns:**

The paper received five reviews, consisting of three generally positive evaluations, one leaning negative, and one negative assessment. In response, the authors significantly revised the manuscript, incorporated new baselines, extended model scalability experiments, and provided clarifications on the data processing pipelines. Given that baseline comparisons and scalability were the primary concerns raised by the reviewers, this targeted rebuttal is well-positioned to improve the final scores.

**Reviewer Scores:**

Because the revisions directly tackle the reviewers' most serious concerns regarding scalability and baselines, there is a strong potential for a positive score adjustment.

---

### Decision · Program_Chairs · 2026-01-26

Accept (Poster)